# Nitrogen Management of Variety Screening of Winter Wheat Cultivated on Chernozem Soil for Yield Optimization

**DOI:** 10.3390/plants14233580

**Published:** 2025-11-24

**Authors:** Elena Rosculete, Ramona Aida Paunescu, Catalin Aurelian Rosculete, Gabriela Paunescu, Elena Bonciu, Aurel Liviu Olaru, Denisa Florența Murtaza

**Affiliations:** 1Department of Land Measurement, Management, Mechanization, Faculty of Agronomy, University of Craiova, 13 A.I. Cuza Street, 200585 Craiova, Romania; elena.rosculete@edu.ucv.ro; 2Department of Agricultural and Forestry Technologies, Faculty of Agronomy, University of Craiova, 13 A.I. Cuza Street, 200585 Craiova, Romania; catalin.rosculete@edu.ucv.ro (C.A.R.); elena.bonciu@edu.ucv.ro (E.B.); aurel.olaru@edu.ucv.ro (A.L.O.); 3SCDA Caracal, University of Craiova, 106 Vasile Alecsandri Street, 235200 Caracal, Romania; gabriela.paunescu@adm.ucv.ro (G.P.);; 4Doctoral School of Animal and Plant Resources Engineering (IRAV), University of Craiova, 13 A.I. Cuza Street, 200585 Craiova, Romania

**Keywords:** nitrate nitrogen, ammonium nitrogen, chernozem soil, yield, wheat

## Abstract

Nitrogen is one of the most essential nutrients for agricultural crops, and optimizing nitrogen fertilization enables the achievement of high yields and improved quality. In this context, the aim of this study was to identify the nitrogen form that influences wheat yield the most significantly, as well as the cultivars that respond positively to specific forms of nitrogen fertilization, in order to provide recommendations regarding cultivar selection and the appropriate technological approach for chernozem soils in southern Romania. Over a period of three agricultural years (2021–2022, 2022–2023, 2023–2024), 36 winter wheat cultivars were tested under three distinct fertilization conditions, nitrate nitrogen, ammonium nitrogen, and nitrate + ammonium nitrogen, each applied at three different rates: 120 kg·ha^−1^ active substance (a.s.), 150 kg·ha^−1^ a.s., and 170 kg·ha^−1^ a.s. The comparative performance of each cultivar relative to the others was evaluated using the Newman–Keuls multiple-range test. The coefficient of variation (CV) of the obtained yields was used to determine yield stability, and its correlation with yield levels allowed for the identification and recommendation of cultivars that simultaneously demonstrated high yields (above average) and good or moderate stability. Sole application of ammonium nitrogen significantly reduced yield by 3.34% (from 70.66 Q/ha to 68.3 Q/ha), while the nitrate+ammonium combination showed a non-significant yield increase compared to nitrate nitrogen alone (+0.65%, *p* > 0.05). Among the tested cultivars, Sacramento was identified as the most productive, showing statistically significant superiority for Ct1—the most commonly grown wheat cultivar Glosa—as well as for control 2 (Ct2), which represented the average yield of all tested cultivars under all nitrogen fertilization treatments.

## 1. Introduction

At the global level, the Earth’s population continues to follow an upward trend, which requires an increase in agricultural crop production in order to avoid food insecurity. However, agricultural intensification has led to environmental issues such as climate change and resource depletion [1].

In the context of this demographic development, ensuring food security remains a pressing issue that can be addressed by maintaining and increasing agricultural productivity—a goal that largely depends on the adoption of new technologies and agricultural practices. Among these, the use of fertilizers plays a pivotal role.

Nitrogen fertilization is a commonly applied agricultural management strategy that positively influences both the yield and quality of crops, including cereals. However, optimizing nitrogen fertilization remains a challenge, as improper nitrogen management can have adverse effects on crops and environmental quality [2]. Nitrogen (N) is a key factor in increasing crop yield, significantly influencing leaf area development, light interception, photosynthetic efficiency, and the accumulation of dry matter in crop plants [3].

Due to the high effectiveness of nitrogen fertilization in enhancing yield, nitrogen fertilizers have been widely recommended in recent years for use in modern wheat production, contributing to increased productivity. This has even led to the common perception that applying more nitrogen results in higher yields [4,5,6].

Nitrogenous fertilizers have contributed to increased yields, especially in major crops such as wheat [7]. N is one of the three essential macronutrients required for plant development and is a key component of many essential plant compounds, including amino acids, nucleotides, proteins, and, particularly, enzymes involved in critical metabolic processes [8]. N is also a major structural element of chlorophyll and is present in certain plant hormones, either directly or as N-containing derivatives [9].

Nitrogen availability is a limiting factor for wheat yield [10], and there is significant variability among wheat cultivars in terms of nitrogen uptake and utilization for yield formation [11]. Therefore, applying the appropriate nitrogen rate is a fundamental means to enhance grain yield, nitrogen uptake, and use efficiency. However, nitrogen losses through volatilization can reduce both yield and nutrient use efficiency in wheat [12,13]. Previous and recent studies have shown that a large portion of granular nitrogen that is applied to the soil may be lost through leaching (nitrate nitrogen) or volatilization (ammonium nitrogen) [14]. As a result, researchers have recommended compensating for these losses through the use of stabilized urea-based nitrogen fertilizers [15,16,17,18].

Ammonium nitrogen (NH_4_^+^-N) and nitrate nitrogen (NO_3_^−^-N) are the main forms of nitrogen absorbed by plants, together comprising approximately 70% of total cation and anion uptake [19]. Ammonia nitrogen, which includes ammonia (NH_3_) and free ammonium ions (NH_4_^+^), is an important chemical component in agriculture and biotechnology, with its speciation depending primarily on the pH of the surrounding environment [20,21]. Although nitrate (NO_3_^−^-N), ammonium (NH_4_^+^-N), and amide (NH_2_-N) forms of nitrogen are used in fertilization, nitrate is the most readily absorbed form by plant roots. The other forms must undergo biochemical transformations in the soil—except in cases where cereals [22] and grasses [23,24] selectively absorb ammonium nitrogen.

Nitrate nitrogen is the most immediately available form of synthetic nitrogen fertilizer for plant roots. Other forms must be converted by soil microorganisms into nitrate before becoming plant-accessible. Because of its rapid mobility, nitrate nitrogen is easily leached from the soil, which is why it is frequently applied as a top dressing during crop development. Numerous studies have investigated the effects of various nitrogen fertilizers on wheat yields, with results differing according to fertilizer form, application timing, and environmental conditions [7,25,26]. Due to its strong efficacy in yield enhancement, nitrogen fertilizer has been broadly recommended for use in wheat production [27,28]. However, excessive nitrogen application drastically reduces the nutrient use efficiency and poses serious environmental and ecological risks [1,2]. High nitrogen inputs result in significant nitrogen losses to groundwater, further diminishing efficiency [29].

Despite wheat’s global importance in terms of cultivated area, geographic distribution, and total output, relatively few studies have investigated how different nitrogen forms are absorbed by this crop [10,30,31]. Both ammonium and nitrate nitrogen are predominant forms of mineral nitrogen in soils; however, nitrate tends to dominate under favorable conditions. Ammonium, whether added directly or released via mineralization of organic matter, is typically rapidly nitrified into nitrate. The ammonium-to-nitrate ratio in soil depends on the nitrification rate, which is inhibited under acidic or anaerobic conditions [32]. Some researchers estimate that global nitrogen consumption would need to reach 236 million tonnes to support the 70–110% increase in cereal production that is required to meet food demands for a projected global population of 8.0 to 10.4 billion by 2050 [33].

High concentrations of ammonia (e.g., above 0.5 µmol/L) can have significant environmental impacts [34]. Although determining the optimal nitrate level in soils remains difficult, reducing its accumulation is considered essential [32,35,36]. In general, wheat seedlings exhibit distinct physiological responses depending on the nitrogen form supplied [37,38,39,40]. Many studies have demonstrated that mixed nitrogen forms often result in superior plant growth indices in wheat, maize, and other species [41,42,43]. Effective nutritional management requires detailed knowledge of the agronomic characteristics of the soils in a given farm and the specific nutrient requirements of the cultivated plants, as well as the ability to identify deficiency symptoms in a timely manner. In winter wheat, nutrient uptake begins to increase significantly in the spring, particularly at the onset of stem elongation [44,45]. Under future climate scenarios characterized by elevated CO_2_ concentrations, the use of nitrification inhibitors may be recommended to delay the conversion of ammonium to nitrate, thus preserving nitrogen in the ammonium form for longer periods in dryland cropping systems, where nitrate dominates [46]. Optimizing nitrogen fertilization not only enhances yield and grain quality but also improves economic returns and reduces environmental pollution [47,48].

The aim of this study was to identify the nitrogen form that influences wheat yield the most significantly, as well as the cultivars that respond positively to specific form of nitrogen fertilization, in order to provide recommendations regarding cultivar selection and the appropriate technological approach for chernozem soils in southern Romania. The originality of the study lies in the comparisons made between several forms of nitrogen, as well as in the multiple comparisons between a large number of wheat cultivars of different origins. These are relatively rare.

## 2. Results and Discussion

### 2.1. Influence of Cultivar (Factor A) on Yield

When analyzing the individual influence of Factor A—cultivar—on yield, based on the difference from control 1 (Ct1)—the most commonly grown wheat cultivar Glosa (72.19 Q (Quintals)/ha)—and from control 2 (Ct2)—the average yield of all tested cultivars (70.03 Q/ha)—the cultivars can be grouped into several categories (Table 1) as follows: cultivars with statistically significant yield increases compared to both controls: Rubisko, Sacramento, Centurion, Activus, Crișana; cultivars with statistically significant yield increases compared to Ct2: Vivendo, Litera; cultivars with statistically significant yield reductions compared to both controls: Combin, Aspekt, Pitar, Dacic, Biharia, Miranda, Alex; cultivars with statistically significant yield reductions compared to Ct1: Tika Taka, Papillon, Boema, Tiberius, Arezzo; and cultivars with no statistically significant differences compared to either control: Apexus, Solindo, Basilio, Chevignon, Sosthene, Sothys, Aurelius, Sophie, Voinic, Ursita, Abund, Sofru, Gabrio, Sphere, Ciprian, Certiva.

The average yields ranged from 84.18 Q/ha for the cultivar Sacramento to 55.08 Q/ha for the cultivar Dacic. The standard error (SE) of the means for 35 degrees of freedom is +/− 1.04. The confidence interval is 70.03 +/− 1.04, with a lower confidence interval of 68.99 and upper confidence interval of 71.07. In Table 1, values marked in yellow represent statistically significant increases, and values marked in pink represent statistically significant decreases.

Other results show that wheat varieties exhibit different yield potentials and responses to environmental conditions, which directly influence overall productivity [49]. While wheat yield is genetically determined, it is significantly affected by climatic conditions during the growing season and by the agricultural technologies employed [50]. Selecting the appropriate variety for a specific region and its prevailing environmental conditions is essential for optimizing both yield and grain quality. Climate change, along with disease resistance, further emphasizes the importance of variety selection. Moreover, adopting wheat variety mixtures can become a major strategy among farmers, potentially reducing the pesticide dependence of current cropping systems [51].

The yields presented above (Table 1) were obtained under varying levels of favorability for wheat cultivation, depending on climatic conditions across the study years. In the 2021–2022 agricultural year, growing conditions were less favorable, with total precipitation below normal levels—364 mm. In contrast, the 2022–2023 season offered favorable conditions for wheat development, while the 2023–2024 season was characterized by moderate-stress conditions for the crop.

### 2.2. The Effect of Nitrogen Form (Factor B) on Yield

Factor B (nitrogen form) significantly influenced wheat yield (Figure 1). When fertilization was carried out exclusively with ammonium nitrogen, the yield decreased significantly, reaching 68.3 Q/ha. Although a slight yield increase was observed in the treatment combining nitrate nitrogen + ammonium nitrogen, the difference was not statistically significant. The standard error (SE) of the means for 35 degrees of freedom for nitrate N dates is +/− 1.24; for ammonium N dates, it is +/− 1.28; and for nitrate + ammonium dates, it is +/− 1.80. For the nitrate N, the confidence interval is 70.66 +/− 2.52, with a lower confidence interval of 68.14 and upper confidence interval of 73.18. For the ammonium N, the confidence interval is 68.67 +/− 2.60, with a lower confidence interval of 66.07 and upper confidence interval of 71.27, and for the nitrate + ammonium treatment, the confidence interval is 70.93 +/− 3.65, with a lower confidence interval of 67.28 and upper confidence interval of 74.58.

Similar findings have been reported in other studies examining the effects of ammonium nitrogen. For instance, a study conducted in Poland [47] revealed that the average yield of winter wheat that was fertilized solely with ammonium nitrogen was 2.7–4.2% lower than the yield obtained following the application of other nitrogen fertilizer forms. In some dryland areas of China, wheat exhibited better responses to nitrate nitrogen in most cases, while in others, no significant difference was found between nitrate and ammonium nitrogen in terms of yield performance [52]. The authors concluded that the superiority of nitrate nitrogen over ammonium nitrogen in influencing wheat yield depends on the cumulative nitrate content in the soil. Specifically, nitrate nitrogen only outperforms ammonium nitrogen in soils with low cumulative nitrate concentrations at the rooting depth [52].

The results of a study examining the effects of different nitrogen fertilizers (urea, ammonium nitrate, ammonium sulfate, and calcium ammonium nitrate) on wheat growth and yield indicated that ammonium nitrate exhibited comparable beneficial effects, while ammonium sulfate and calcium ammonium nitrate had relatively weaker impacts on wheat productivity [7]. At wheat maturity, the correlation between cumulative nitrate nitrogen (0–100 cm soil layer) and total biomass, as well as biomass increase due to nitrogen addition, was the strongest [32]. Another study reported that nitrate nitrogen led to the highest yields and yield increases, followed by the ammonium/nitrate combination in a 1:2 ratio, while ammonium nitrogen alone resulted in the lowest performance [52].

Findings reported in [39] suggest that wheat and maize plants performed better under mixed nitrogen forms than under single-form applications. Specifically, in single-nitrogen treatments, wheat seedlings supplied with nitrate-N demonstrated better growth than those receiving ammonium-N. Plants receiving only ammonium-N showed low plant height and fewer tillers, whereas those supplied with nitrate-N were characterized by slightly reduced height, but more tillers and greater aboveground biomass. Moreover, under elevated CO_2_ (eCO_2_) conditions, wheat seedlings receiving mixed N supply exhibited a significant increase in carbon concentration in root exudates and a relatively lower nitrogen concentration [39].

Field and pot experiments investigating the effects of ammonium and nitrate on wheat yield consistently showed that nitrate application or nitrate–ammonium combinations led to higher wheat yields compared to ammonium alone [53,54]. However, ammonium had a positive effect on wheat root activity and improved the nitrogen recovery rate [55].

Other authors have reported that applying ammonium or nitrate nitrogen alone may induce pH changes in the growth medium, leading to cation–anion imbalances, whereas their simultaneous application can regulate both cell and rhizosphere pH [56,57]. The combined application not only helps maintain the availability of nutrients, such as phosphorus and micronutrients, but also contributes to protecting the soil environment [57].

In general, regarding the morphological development of wheat, ammonium-N supply primarily affected maximum root length and tiller number. The negative effect of ammonium-N on root length was found to be more pronounced under ambient CO_2_ (aCO_2_) conditions [39].

Additionally, recent studies have highlighted the critical role of nitrogen forms in modulating host–pathogen interactions and shaping the wheat rhizosphere microbiome composition [58].

### 2.3. The Effect of Nitrogen Dose (Factor C) on Yield

The applied nitrogen dose (Factor C) had a substantial impact on yield (Figure 2). Higher doses resulted in yields that were significantly lower than the yield obtained with the 120 kg·ha^−1^ a.s. dose, regardless of the nitrogen form used.

Future research will focus on the inclusion of physiological indicators such as chlorophyll content. The lack of these indicators, as well as the lack of determination of plant nitrogen content or biomass allocation, represents a limitation of this study.

The results are consistent with those reported by other authors, who have shown that high nitrogen doses and excessive nitrogen application lead to reduced yield and increased nitrogen losses in the wheat–soil system [59,60]. Recent studies have indicated that a dose of 180 kg N ha^−1^ was the most effective for improving wheat growth, physiological efficiency, and grain yield, with 135 kg N ha^−1^ also showing favorable results. In contrast, higher doses (225 and 270 kg N ha^−1^) resulted in diminished performance, suggesting a threshold beyond which nitrogen becomes counterproductive [60]. Similarly, increasing nitrogen application from 70 to 130 kg ha^−1^ had a positive effect on wheat yield in experiments conducted in Poland on rendzina soils [61].

On the other hand, according to [7], urea applied at 240 kg·ha^−1^ resulted in the tallest plants (92.5 cm), followed by ammonium nitrate at the same rate (88.4 cm). Furthermore, urea at 240 kg·ha^−1^ led to a grain yield of 5300 kg·ha^−1^, while ammonium nitrate achieved a slightly higher yield of 5400 kg·ha^−1^ at the same application rate [7].

Previous research has emphasized the importance of optimizing nitrogen management in wheat cultivation. Strategies such as split nitrogen application at different growth stages have been shown to improve yield while minimizing environmental risks, including nitrogen losses, water contamination, and greenhouse gas emissions [62,63].

In contrast to nitrate N, ammonium N has been shown to exhibit toxicity to plants [53]. Visibly, this toxicity manifests through inhibited growth, reduced leaf area and biomass, delayed and restricted root development, and the formation of fine or dark-colored roots [30].

Optimizing nitrogen fertilization not only enables high and high-quality yields, but also ensures economic profitability and contributes to environmental protection [47,48]. The search for strategies to help farmers optimize nitrogen fertilizer use is of global importance [64]. Nitrogen ranks second only to precipitation as the most frequent limiting factor for yield; when nitrogen supplied to wheat is not used efficiently, it may be lost from the cropping system to the surrounding environment [65].

### 2.4. Influence of the Cultivar × Nitrogen Form Interaction (Factor A × Factor B)

The interaction between the cultivar and the form of nitrogen fertilizer applied is of particular significance. Its impact is considerable, as cultivars exhibit differentiated responses. The wide range of cultivars tested revealed remarkable variability in their behavior depending on the nitrogen form used compared both to the reference cultivar Glosa (Ct1) and to the mean performance of all cultivars (Ct2). In the experimental design, cultivars were denoted as C = cultivar. These findings provide valuable guidance, especially in practical scenarios where a specific form of nitrogen is available, and the objective is to select the most suitable cultivar for optimal performance.

Among the cultivars tested, three were distinguished by their stable performance, demonstrating equivalence with the control in all three nitrogen treatments (Figure 3). In other words, the cultivars Sosthene (Figure 3a), Ciprian (Figure 3b), and Certiva (Figure 3c) exhibited comparable productivity to both Glosa and the average of the 36 wheat cultivars assessed, across all fertilization regimes: nitrate nitrogen (Magnisal), ammonium nitrogen (ammonium sulfate), or combined nitrate and ammonium nitrogen (ammonium nitrate).

Although ammonium nitrogen did not provide the most favorable conditions for fully expressing yield potential, several cultivars—including Vivendo (Figure 4a), Litera (Figure 4b), Centurion (Figure 4c), Activus (Figure 4d), Sophie (Figure 4e), and Ursita (Figure 4f)—demonstrated significantly higher productivity than both the reference cultivar Glosa and the overall mean under this nitrogen form. The cultivar Tiberius also showed superior performance, although only in comparison with the general average (Figure 5).

Overall, the most productive cultivar was Sacramento (Figure 6a), which demonstrated significantly higher yields than both controls under all nitrogen fertilization regimes. A similar trend was observed for Rubisko (Figure 6b), although this superiority was not evident under ammonium nitrogen fertilization.

At the opposite end of the spectrum, cultivars with inferior productivity compared to both reference points, regardless of the nitrogen form applied, were Combin (Figure 7a) and Dacic (Figure 7b).

The cultivars Tika Taka and Pitar were inferior only to Glosa across all nitrogen fertilizer forms (Figure 8a,b).

The cultivars Aurelius, Aspekt, Sphere, Miranda, Alex, and Arezzo were unable to efficiently utilize ammonium nitrogen, with significant yield reductions compared to both reference controls (Figure 9a–g).

Among the tested cultivars, Papillon, Solindo, Chevignon, Aurelius, Vivendo, Centurion, Activus, Sphere, and Crișana responded very favorably to ammonium nitrate (a combination of nitrate and ammonium nitrogen in equal parts), achieving significantly higher yields than both controls. Some of these cultivars are graphically represented in Figure 10a–d.

The calculation of the coefficient of variation (CV%) for all yield values obtained from each cultivar—whether grown under fertilization with nitrate nitrogen, ammonium nitrogen, or both forms across all tested doses—allowed us to highlight variants based on their stability and, implicitly, the degree to which they were influenced by the form of nitrogen fertilizer applied. Accordingly, the results suggested that the cultivars Certiva, Litera, and Boema were stable (coefficient of variation below 10%), indicating a minimal influence of the nitrogen form on their performance.

In contrast, the cultivars Aurelius, Sphere, Voinic, Arezzo, Pitar, Biharia, Dacic, and Miranda exhibited coefficients of variation exceeding 20% (indicating instability), which reflects a substantial influence of both the form of nitrogen fertilizer and the applied doses (Figure 11).

Similar findings on the stability of wheat cultivars have been reported by other researchers. Analysis of variation coefficients showed that, regardless of wheat species, cultivars were stable under fertilization with 150 kg N·ha^−1^, with CV values ranging between 4.81% and 6.18% [61]. However, variation exceeding 20% (instability) was observed for certain traits such as gluten elasticity, grain vitreousness, and total ash content [61].

The results concerning the correlation between yield and the variability index revealed a determination coefficient (R^2^) of 29.3%. In the quadrant representing the desired interaction—cultivars with high yields and low variability indexes, defined by the mean values of these two traits—Sacramento, Vivendo, Centurion, Litera, and Certiva stood out under the pedo-climatic conditions of Caracal (Figure 12). The yields were represented as the average values obtained across the three nitrogen fertilization variants, differentiated by nitrogen form.

The Newman–Keuls test confirmed several results demonstrating the extent to which one cultivar is more productive compared to another winter wheat cultivar tested on the Chernozem soil at Caracal. Thus, the cultivar Sacramento outperformed the cultivars Dacic, Pitar, Aspekt, Miranda, Combin, Alex, Biharia, Boema, Tika Taka, Papillon, Arezzo, Tiberius, Aurelius, Abund, and Sphere, regardless of the nitrogen form and dose applied (Table 2). The values marked in green are those that differ significantly from the variety to which they refer (the one specified at the top of the table).

The cultivar Rubisko was superior to Dacic, Pitar, Aspekt, Miranda, Combin, and Alex under the same testing conditions. Similarly, Activus showed higher productivity compared to Dacic, Pitar, Aspekt, Miranda, and Combin. The cultivar Crișana also exceeded the productivity of the aforementioned cultivars. Additionally, Vivendo, and Centurion demonstrated comparable superiority.

The cultivar Litera performed better than Dacic, Pitar, and Aspekt. Conversely, Dacic was inferior in yield compared to several other cultivars, namely, Chevignon, Ciprian, Sophie, Certiva, Solindo, Sothys, Glosa, Ursita, Apexus, Gabrio, Sosthene, and Basilio.

All other tested cultivars not previously highlighted were at a similar production level, indicating the diversity available when selecting cultivars.

Similar findings were reported by other authors who tested multiple durum wheat cultivars [66]. The Newman–Keuls test identified four significantly different groups regarding productivity. The cultivar GTA in the first group was characterized by the highest average yield of 5.46 ± 0.22 t·ha^−1^, while the cultivar Wahbi in the fourth group ranked last, with the lowest grain yield of 3.32 ± 0.16 t·ha^−1^ [66]. According to other studies, the Newman–Keuls test revealed two distinct homogeneous groups, with group (a) containing the highest averages favoring conventional tillage, while no-tillage recorded the lowest averages in group (b) [67].

## 3. Materials and Methods

During three agricultural years (2021–2022, 2022–2023, 2023–2024), 36 winter wheat cultivars were tested under field conditions on Chernozem soil at Caracal, Romania. The experiment involved three different nitrogen fertilization regimes: nitrate nitrogen, ammonium nitrogen, and a combination of nitrate + ammonium nitrogen. Each nitrogen form was applied at three different doses: 120 kg·ha^−1^ a.s., 150 kg·ha^−1^ a.s., and 170 kg·ha^−1^ a.s. The three nitrogen forms were supplied through the following fertilizers: Magnisal (containing 11% nitrate nitrogen), ammonium sulfate (21% ammonium nitrogen), and ammonium nitrate (33.5% total nitrogen: 16.75% nitrate nitrogen + 16.75% ammonium nitrogen).

The experiment was conducted in subdivided plots with three replicates for each of the three-factor interactions: cultivar × nitrogen form × nitrogen dose. The cultivars represented the large plots, the nitrogen forms represented the medium plots, and the doses represented the small plots. First, the variability of the totals of the large plots occupied by Factor A was analyzed, which was determined by the action of Factor A, the differences between replicates, and the random fluctuation corresponding to the large plots. Then, the overall variability of the medium plots occupied by Factor B was analyzed, which also includes the influence of Factor B, the interaction of A × B, and the random fluctuation corresponding to the medium plots. Lastly, the variability of the small plots was studied, which in fact represents the total variability of the experiment.

The experiment was conducted on a typical argic Chernozem (non-calcareous), characterized by a well-defined profile and insignificant variability in physical, hydrological, and chemical properties. This deep soil, with a loam to clay loam texture, developed from aeolian deposits (loess and loess-like carbonate materials). The texture is silty clay loam in the upper 70 cm and silty loam below, resulting in a medium bulk density throughout the soil profile.

The soil profile is characterized by the sequence of the following horizons: Ap, Am, A/B, Bt1, Bt2, B/C, Cca.

Ap = horizon soil A, which represents a soil layer disturbed by plowing or other mechanical action, less than 50 cm thick. The symbol p comes from “processing”, and refers to the fact that in soils taken for cultivation, the upper part of the A horizon (the first 20–25 cm) has a granular structure—a characteristic of plowing.

Am = horizon soil A, molic that is rich in humus of the calcic mull type.

B = horizon soil B, a mineral horizon soil, formed in the lower part of the soil profile, below a horizon A.

Cca = horizon soil C, rich in calcium carbonate.

Total porosity (TP) is medium in Ap (48.8%) and high in Am (58.2%). Air porosity (AP) is medium in Ap (15.2%), high in Am (23.8%), medium in B (21.7%), and low in Cca (12.2%). The variation limits for clay are between 33.4 and 40.3%. The Ap tilled horizon has a clay content of 36%, in contrast to the Am horizon with a content of 40.3%. In the B/C and Cca horizons, the amount of clay is lower (33.8% and 33.4%, respectively). The soil is weakly to moderately compacted on the horizons subadjacent to the tilled horizon. The values of the hydrophysical indices show different values, correlated with the physical properties. There is a medium value for the wilting coefficient in the Ap and Am horizons (11.7%), a high value in B/C (13.2%), and a medium value in Cca (12%).

Field capacity (FC) is average throughout the soil profile, with values of 23.3% in Ap, Am, and B/C and 22.4% in Cca. The soil has a humus content of 3.04% in the Ap horizon and 1.78% in the Am horizon, decreasing by 1.24% in the B horizon. It is moderately supplied with total nitrogen, with 0.172% in Ap, very well supplied with mobile phosphorus (168 ppm) in Ap, and well supplied with mobile potassium (248 ppm). The soil reaction on the profile is slightly acidic, slightly alkaline. The sum of exchangeable bases is medium (23.2 in Ap, 24.8 in Am), and the degree of base saturation is 92.62%, gradually increasing in the Cca horizon. The microelement content is high for zinc (176 ppm) in Ap and extremely low in the B horizon (0.40 ppm).

Meteorological data, based on an 79-year average, indicate a mean annual temperature of 11.1 °C (0.6 °C in winter, 10.8 °C in spring, 22 °C in summer, and 11.4 °C in autumn) and an average precipitation of 389.5 mm during the wheat vegetation period (October–June).

During the three experimental years, rainfall varied considerably:In the 2021–2022 agricultural year, conditions were unfavorable, with below-average precipitation (364 mm).In 2022–2023, total precipitation reached 464.6 mm, supporting favorable wheat development, although the growing season was accompanied by high temperatures, with maximum values reaching 38.7 °C.In 2023–2024, water stress was present, but mitigated by abundant rainfall in May, during the grain-filling phase.

The data was collected by the PESSL automatic weather station installed in the experimental field where the varieties were tested. Its sensors record data on precipitation, temperature, humidity, wind speed, solar radiation, ETO, and dew point.

Table 3 presents the technological data on the management applied to the wheat crop in the experimental set up in Caracal, Romania.

To process the data obtained, we used the PS3F program (3-factor polyfactorial experiment), the Newman–Keuls test for multiple comparisons, the calculation of the coefficient of variability to highlight yield stability, and the correlation coefficient for the relationship between yield and the coefficient of variability. Statistical interpretation of results was based on raport at values *p* < 0.05, *p* < 0.01, and *p* < 0.001 for a three-factorial experimental design, where Factor A was the cultivar, Factor B the nitrogen form, and Factor C the nitrogen dose.

The influence of each individual factor (cultivar, nitrogen form, and dose) on yield was assessed, as well as the interaction effect between cultivar and nitrogen type, using two controls for comparison: Control 1 (Ct1), represented by the cultivar Glosa, and Control 2 (Ct2), represented by the average of all tested cultivars. A result was considered significant at *p* < 0.05, with the corresponding threshold value (LSD) being 5.64 Q(Quintals)/ha for the Chernozem soil at Caracal.

The relative performance of each cultivar in comparison to all others was assessed using the Newman–Keuls test. The coefficient of variation (CV%) of yield, regardless of nitrogen form or dose, was used to assess yield stability. Its correlation with productivity enabled the identification of cultivars that simultaneously exhibited high yields (above average) and good-to-moderate stability, suitable for recommendation.

## 4. Conclusions

The analysis of individual factors—as well as the interaction between the first two (cultivar and nitrogen form)—highlighted their strong influence on yield performance. Yield variability was significant, ranging from 84.18 Q/ha in the Sacramento cultivar to 55.08 Q/ha in the Dacic cultivar.

Sole application of ammonium nitrogen significantly reduced yield by 3.34% (from 70.66 Q/ha to 68.3 Q/ha), while the nitrate+ammonium combination showed a non-significant yield increase compared to nitrate nitrogen alone (+0.65%, *p* > 0.05).

Consistent with numerous global studies, the results obtained on the Chernozem soil at Caracal showed that when fertilization was performed only with ammonium nitrogen, yields declined significantly. In the variant fertilized with ammonium nitrate (a combination of nitrate and ammonium nitrogen), a slight increase in yield was observed, although it was not statistically significant compared to fertilization with nitrate nitrogen alone.

Among the wide assortment of tested common winter wheat cultivars (36 cultivars of diverse origin), only one cultivar—Sacramento—was significantly superior to both controls under all nitrogen forms evaluated.

The optimal recommended nitrogen dose, regardless of nitrogen form, was 120 kg a.s./ha. Yields obtained at the higher tested doses were significantly lower.

## Figures and Tables

**Figure 1 plants-14-03580-f001:**
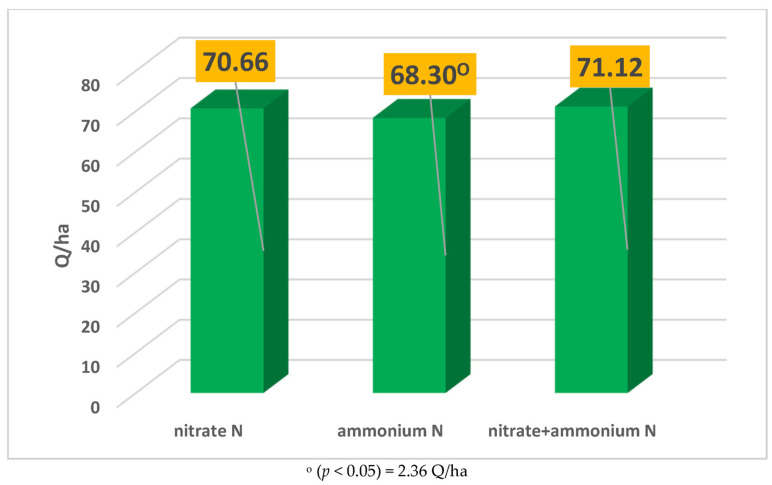
The effect of nitrogen form (Factor B) on yield.

**Figure 2 plants-14-03580-f002:**
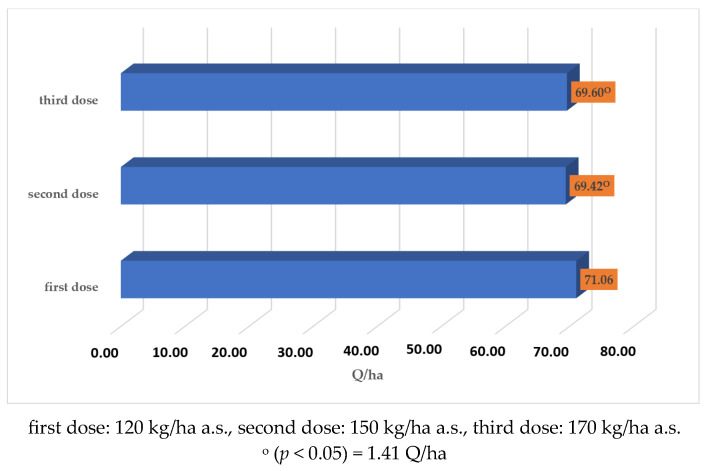
The effect of Factor C (nitrogen dose) on wheat yield.

**Figure 3 plants-14-03580-f003:**
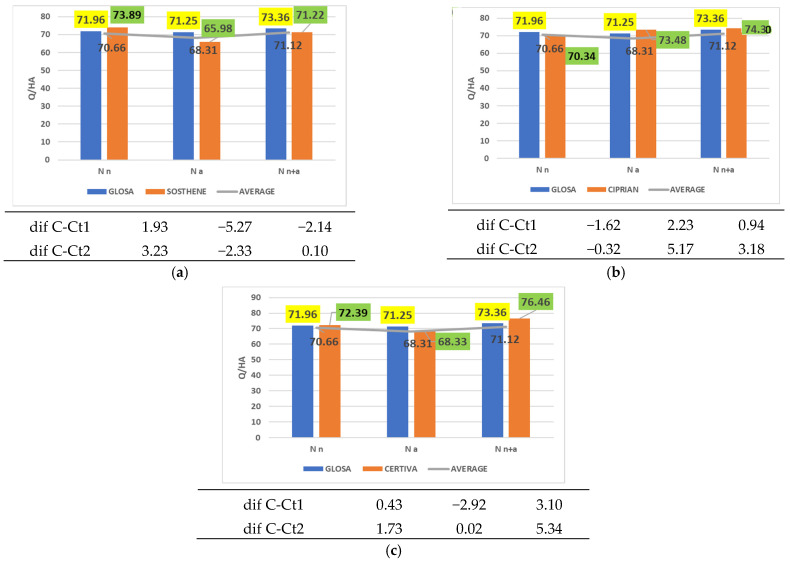
Results regarding the interaction between cultivar and nitrogen form applied. (**a**) Performance of cultivar Sosthene; (**b**) performance of cultivar Ciprian; (**c**) performance of cultivar Certiva.

**Figure 4 plants-14-03580-f004:**
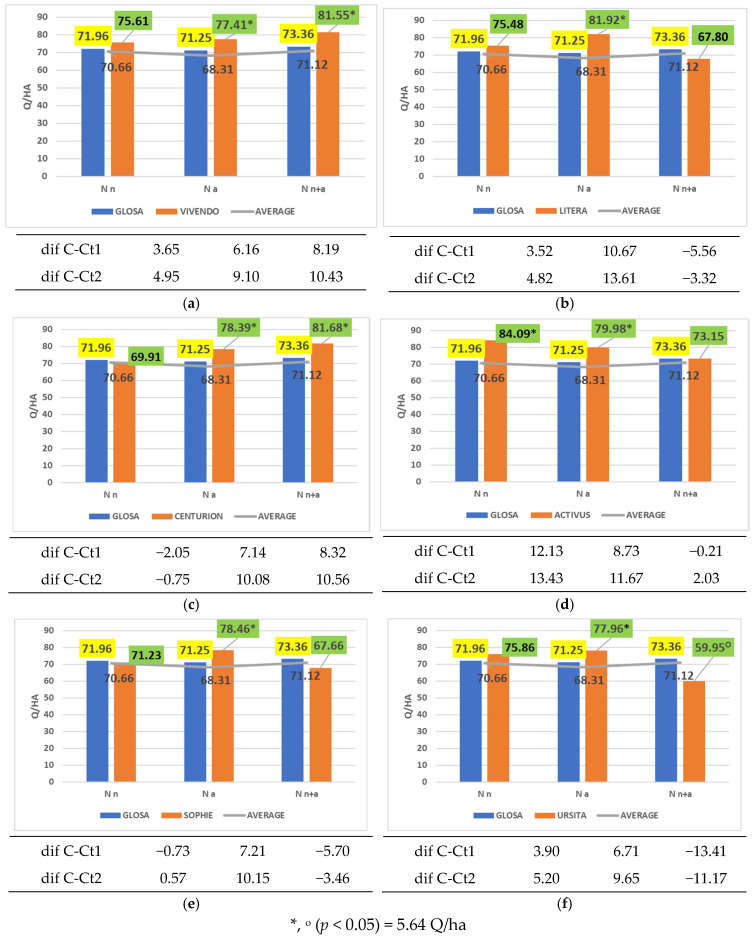
Results regarding the interaction between cultivar and form of nitrogen fertilizer applied. (**a**) Performance of cultivar Vivendo; (**b**) performance of cultivar Litera; (**c**) performance of cultivar Centurion; (**d**) performance of cultivar Activus; (**e**) performance of cultivar Sophie; (**f**) performance of cultivar Ursita.

**Figure 5 plants-14-03580-f005:**
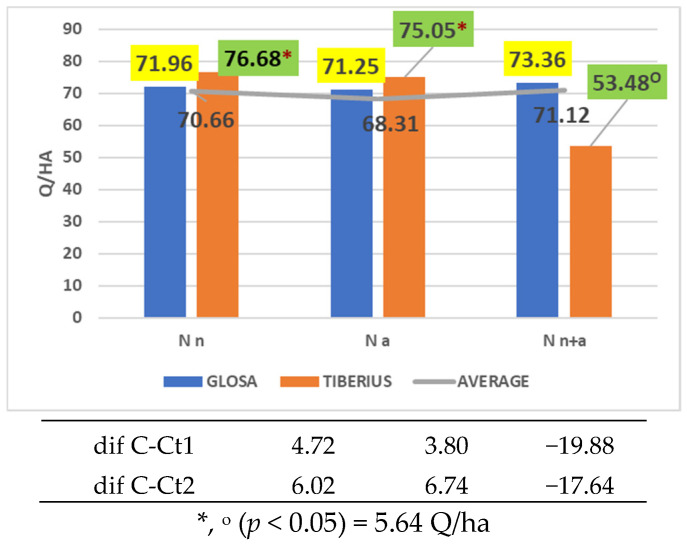
Results regarding the cultivar × nitrogen form interaction for the Tiberius cultivar.

**Figure 6 plants-14-03580-f006:**
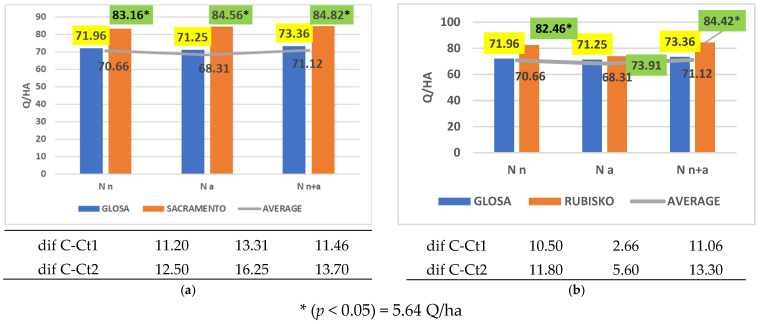
Results regarding the interaction between cultivar and nitrogen form applied. (**a**) Performance of cultivar Sacramento; (**b**) performance of cultivar Rubisko.

**Figure 7 plants-14-03580-f007:**
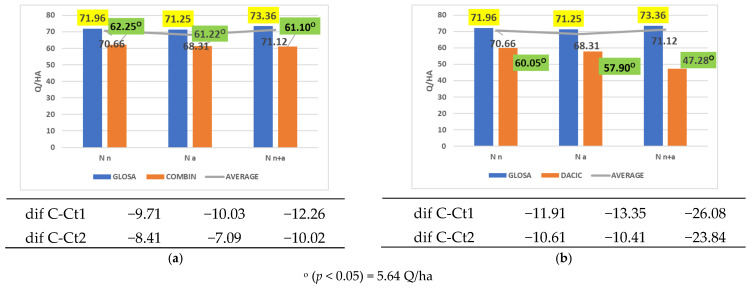
Results regarding the interaction between cultivar and nitrogen form applied. (**a**) Performance of cultivar Combin; (**b**) performance of cultivar Dacic.

**Figure 8 plants-14-03580-f008:**
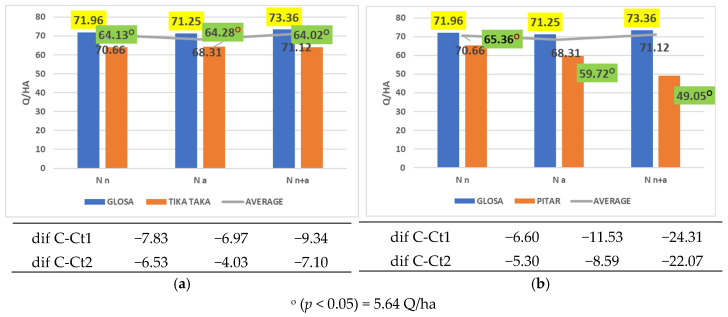
Results regarding the interaction between cultivar and nitrogen form applied. (**a**) Performance of cultivar Tika Taka; (**b**) performance of cultivar Pitar.

**Figure 9 plants-14-03580-f009:**
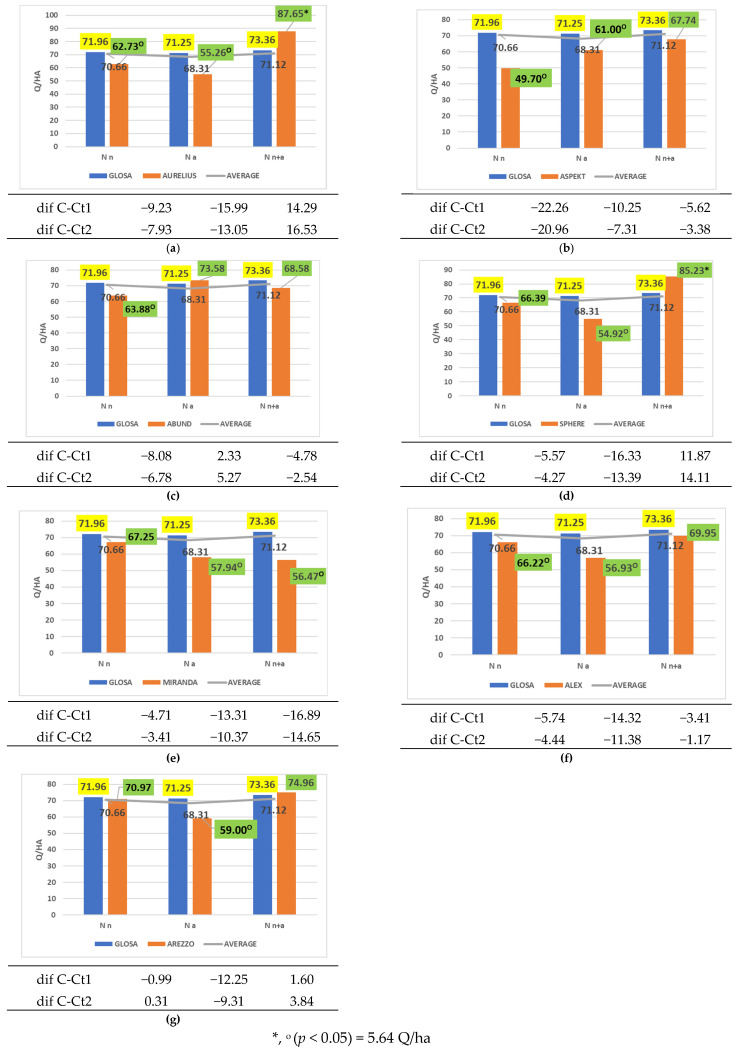
Results regarding the interaction between cultivar and form of nitrogen fertilizer applied. (**a**) Performance of cultivar Aurelius; (**b**) performance of cultivar Aspekt; (**c**) performance of cultivar Abund; (**d**) performance of cultivar Sphere; (**e**) performance of cultivar Miranda; (**f**) performance of cultivar Alex; (**g**) performance of cultivar Arezzo.

**Figure 10 plants-14-03580-f010:**
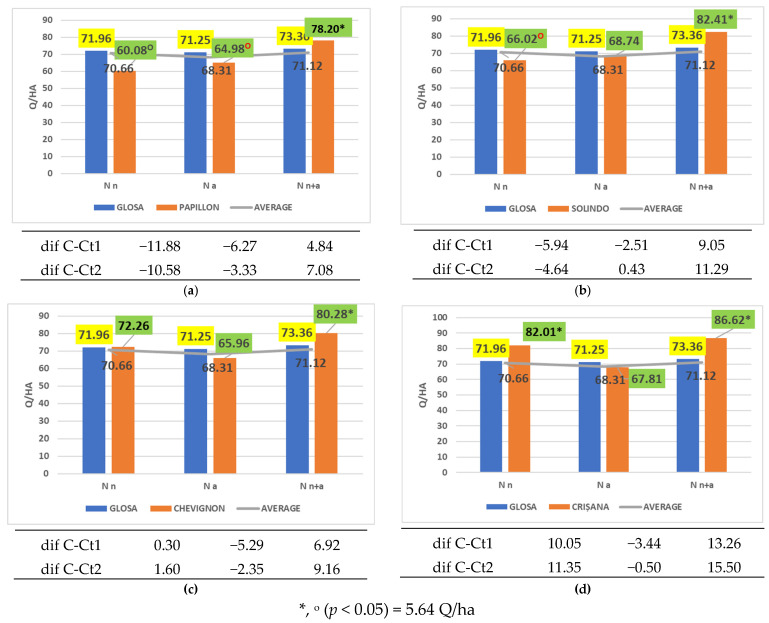
Results regarding the interaction between cultivar and nitrogen form applied. (**a**) Performance of cultivar Papillon; (**b**) performance of cultivar Solindo; (**c**) performance of cultivar Chevignon; (**d**) performance of cultivar Crișana.

**Figure 11 plants-14-03580-f011:**
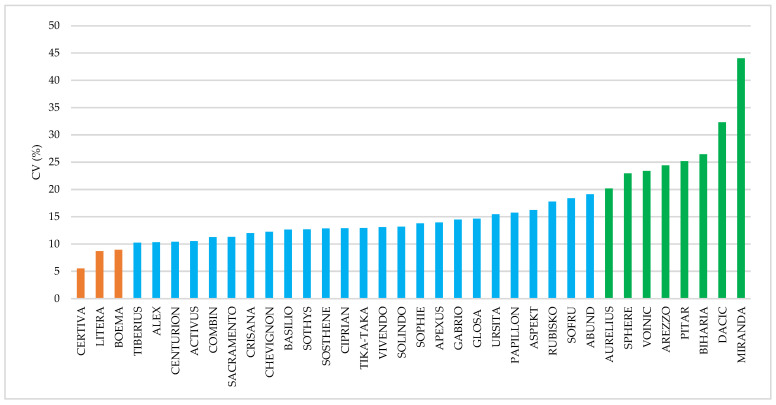
Results regarding the stability of the tested cultivars.

**Figure 12 plants-14-03580-f012:**
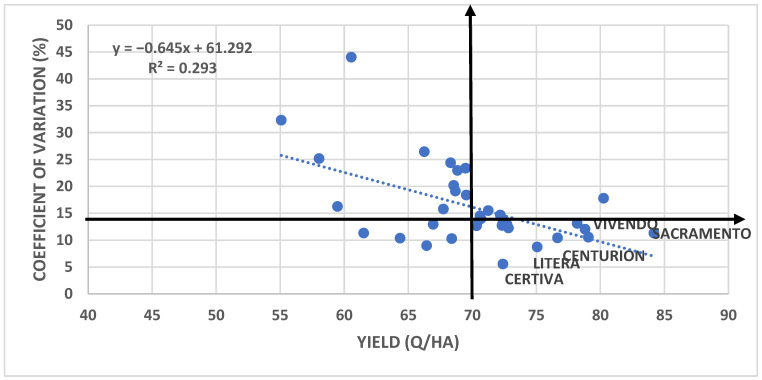
Correlation between yield and variability index of the tested cultivars.

**Table 1 plants-14-03580-t001:** The individual effect of Factor A—cultivar—on wheat yield.

	Cultivars	Yield Q/ha	Difference Ct1	Significance	Difference Ct2	Significance
1	**GLOSA(Ct1)**	72.19	0.00		2.16	
2	**RUBISKO**	80.26	8.07	***	10.23	***
3	**APEXUS**	70.68	−1.51		0.65	
4	**TIKATAKA**	66.95	−5.24	o	−3.08	
5	**PAPILLON**	67.75	−4.44	o	−2.28	
6	**SOLINDO**	72.39	0.20		2.36	
7	**COMBIN**	61.52	−10.67	ooo	−8.51	ooo
8	**BASILIO**	70.34	−1.85		0.31	
9	**CHEVIGNON**	72.83	0.64		2.80	
10	**VIVENDO**	78.19	6.00		8.16	***
11	**SOSTHENE**	70.36	−1.83		0.33	
12	**LITERA**	75.07	2.88		5.04	*
13	**BOEMA**	66.44	−5.75	oo	−3.59	
14	**SOTHYS**	72.32	0.13		2.29	
15	**SACRAMENTO**	84.18	11.99	***	14.15	***
16	**AURELIUS**	68.55	−3.64		−1.48	
17	**ASPEKT**	59.48	−12.71	ooo	−10.55	ooo
18	**CENTURION**	76.66	4.47	*	6.63	*
19	**ACTIVUS**	79.07	6.88	**	9.04	***
20	**SOPHIE**	72.45	0.26		2.42	
21	**TIBERIUS**	68.40	−3.79	o	−1.63	
22	**VOINIC**	69.48	−2.71		−0.55	
23	**URSITA**	71.25	−0.94		1.22	
24	**ABUND**	68.68	−3.51		−1.35	
25	**SOFRU**	69.53	−2.66		−0.50	
26	**GABRIO**	70.61	−1.58		0.58	
27	**PITAR**	58.04	−14.15	ooo	−11.99	ooo
28	**SPHERE**	68.85	−3.34		−1.18	
29	**DACIC**	55.08	−17.11	ooo	−14.95	ooo
30	**BIHARIA**	66.26	−5.93	oo	−3.77	o
31	**MIRANDA**	60.55	−11.64	ooo	−9.48	ooo
32	**ALEX**	64.37	−7.82	ooo	−5.66	oo
33	**CIPRIAN**	72.71	0.52		2.68	
34	**CRISANA**	78.81	6.62	**	8.78	***
35	**CERTIVA**	72.39	0.20		2.36	
36	**AREZZO**	68.31	−3.88	o	−1.72	
	**Average(Ct2)**	**70.03**				

*, ^o^ (*p* < 0.05) = 3.74 Q/ha; **, ^oo^ (*p* < 0.01) = 5.31 Q/ha; ***^, ooo^ (*p* < 0.001) = 7.69 Q/ha.

**Table 2 plants-14-03580-t002:** Results of the Newman–Keuls test showing how the yield of each cultivar compares with all other cultivars.

Sample	Sample Value	Control	Wheat Varieties	Q/ha	DACIC	PITAR	ASPEKT	MIRANDA
		C 36	DACIC	55.08				
Q2.72	12.03	C 35	PITAR	58.04	2.96			
Q3.72	12.67	C 34	ASPEKT	59.48	4.40	1.44		
Q4.72	13.05	C 33	MIRANDA	61.10	6.02	3.06	1.62	
Q5.72	13.35	C 32	COMBIN	61.52	6.44	3.48	2.04	0.42
Q6.72	13.60	C 31	ALEX	64.37	9.29	6.33	4.89	3.27
Q7.72	13.77	C 30	BIHARIA	66.26	11.18	8.22	6.78	5.16
Q8.72	13.94	C 29	BOEMA	66.44	11.36	8.40	6.96	5.34
Q9.72	14.07	C 28	TIKATAKA	66.95	11.87	8.91	7.47	5.85
Q10.72	14.15	C 27	PAPILLON	67.75	12.67	9.71	8.27	6.65
Q11.72	14.15	C 26	AREZZO	68.31	13.23	10.27	8.83	7.21
Q12.72	14.32	C 25	TIBERIUS	68.40	13.32	10.36	8.92	7.30
Q13.72	14.32	C 24	AURELIUS	68.55	13.47	10.51	9.07	7.45
Q14.72	14.49	C 23	ABUND	68.68	13.60	10.64	9.20	7.58
Q15.72	14.49	C 22	SPHERE	68.85	13.77	10.81	9.37	7.75
Q16.72	14.58	C 21	VOINIC	69.48	14.40	11.44	10.00	8.38
Q17.72	14.58	C 20	SOFRU	69.53	14.45	11.49	10.05	8.43
Q18.72	14.66	C 19	BASILIO	70.34	15.26	12.30	10.86	9.24
Q19.72	14.66	C 18	SOSTHENE	70.36	15.28	12.32	10.88	9.26
Q20.72	14.75	C 17	GABRIO	70.61	15.53	12.57	11.13	9.51
Q21.72	14.75	C 16	APEXUS	70.67	15.59	12.63	11.19	9.57
Q22.72	15.05	C 15	URSITA	71.26	16.18	13.22	11.78	10.16
Q23.72	15.05	C 14	GLOSA	72.19	17.11	14.15	12.71	11.09
Q24.72	15.05	C 13	SOTHYS	72.32	17.24	14.28	12.84	11.22
Q25.72	15.05	C 12	SOLINDO	72.39	17.31	14.35	12.91	11.29
Q26.72	15.05	C 11	CERTIVA	72.39	17.31	14.35	12.91	11.29
Q27.72	15.05	C 10	SOPHIE	72.45	17.37	14.41	12.97	11.35
Q28.72	15.05	C 9	CIPRIAN	72.71	17.63	14.67	13.23	11.61
Q29.72	15.05	C 8	CHEVIGNON	72.83	17.75	14.79	13.35	11.73
Q30.72	15.05	C 7	LITERA	75.07	19.99	17.03	15.59	13.97
Q31.72	15.05	C 6	CENTURION	76.66	21.58	18.62	17.18	15.56
Q32.72	15.05	C 5	VIVENDO	78.19	23.11	20.15	18.71	17.09
Q33.72	15.05	C 4	CRISANA	78.81	23.73	20.77	19.33	17.71
Q34.72	15.05	C 3	ACTIVUS	79.07	23.99	21.03	19.59	17.97
Q35.72	15.05	C 2	RUBISKO	80.26	25.18	22.22	20.78	19.16
Q36.72	15.05	C 1	SACRAMENTO	84.18	29.10	26.14	24.70	23.08
Sample	Sample value	Control	Wheat varieties	Q/ha	COMBIN	ALEX	BIHARIA	BOEMA
		C 36	DACIC	55.08				
Q2.72	12.03	C 35	PITAR	58.04				
Q3.72	12.67	C 34	ASPEKT	59.48				
Q4.72	13.05	C 33	MIRANDA	61.10				
Q5.72	13.35	C 32	COMBIN	61.52				
Q6.72	13.60	C 31	ALEX	64.37	2.85			
Q7.72	13.77	C 30	BIHARIA	66.26	4.74	1.89		
Q8.72	13.94	C 29	BOEMA	66.44	4.92	2.07	0.18	
Q9.72	14.07	C 28	TIKATAKA	66.95	5.43	2.58	0.69	0.51
Q10.72	14.15	C 27	PAPILLON	67.75	6.23	3.38	1.49	1.31
Q11.72	14.15	C 26	AREZZO	68.31	6.79	3.94	2.05	1.87
Q12.72	14.32	C 25	TIBERIUS	68.40	6.88	4.03	2.14	1.96
Q13.72	14.32	C 24	AURELIUS	68.55	7.03	4.18	2.29	2.11
Q14.72	14.49	C 23	ABUND	68.68	7.16	4.31	2.42	2.24
Q15.72	14.49	C 22	SPHERE	68.85	7.33	4.48	2.59	2.41
Q16.72	14.58	C 21	VOINIC	69.48	7.96	5.11	3.22	3.04
Q17.72	14.58	C 20	SOFRU	69.53	8.01	5.16	3.27	3.09
Q18.72	14.66	C 19	BASILIO	70.34	8.82	5.97	4.08	3.90
Q19.72	14.66	C 18	SOSTHENE	70.36	8.84	5.99	4.10	3.92
Q20.72	14.75	C 17	GABRIO	70.61	9.09	6.24	4.35	4.17
Q21.72	14.75	C 16	APEXUS	70.67	9.15	6.30	4.41	4.23
Q22.72	15.05	C 15	URSITA	71.26	9.74	6.89	5.00	4.82
Q23.72	15.05	C 14	GLOSA	72.19	10.67	7.82	5.93	5.75
Q24.72	15.05	C 13	SOTHYS	72.32	10.80	7.95	6.06	5.88
Q25.72	15.05	C 12	SOLINDO	72.39	10.87	8.02	6.13	5.95
Q26.72	15.05	C 11	CERTIVA	72.39	10.87	8.02	6.13	5.95
Q27.72	15.05	C 10	SOPHIE	72.45	10.93	8.08	6.19	6.01
Q28.72	15.05	C 9	CIPRIAN	72.71	11.19	8.34	6.45	6.27
Q29.72	15.05	C 8	CHEVIGNON	72.83	11.31	8.46	6.57	6.39
Q30.72	15.05	C 7	LITERA	75.07	13.55	10.70	8.81	8.63
Q31.72	15.05	C 6	CENTURION	76.66	15.14	12.29	10.40	10.22
Q32.72	15.05	C 5	VIVENDO	78.19	16.67	13.82	11.93	11.75
Q33.72	15.05	C 4	CRISANA	78.81	17.29	14.44	12.55	12.37
Q34.72	15.05	C 3	ACTIVUS	79.07	17.55	14.70	12.81	12.63
Q35.72	15.05	C 2	RUBISKO	80.26	18.74	15.89	14.00	13.82
Q36.72	15.05	C 1	SACRAMENTO	84.18	22.66	19.81	17.92	17.74
Sample	Sample value	Control	Wheat varieties	Q/ha	TIKATAKA	PAPILLON	AREZZO	TIBERIUS
		C 36	DACIC	55.08				
Q2.72	12.03	C 35	PITAR	58.04				
Q3.72	12.67	C 34	ASPEKT	59.48				
Q4.72	13.05	C 33	MIRANDA	61.10				
Q5.72	13.35	C 32	COMBIN	61.52				
Q6.72	13.60	C 31	ALEX	64.37				
Q7.72	13.77	C 30	BIHARIA	66.26				
Q8.72	13.94	C 29	BOEMA	66.44				
Q9.72	14.07	C 28	TIKATAKA	66.95				
Q10.72	14.15	C 27	PAPILLON	67.75	0.80			
Q11.72	14.15	C 26	AREZZO	68.31	1.36	0.56		
Q12.72	14.32	C 25	TIBERIUS	68.40	1.45	0.65	0.09	
Q13.72	14.32	C 24	AURELIUS	68.55	1.60	0.80	0.24	0.15
Q14.72	14.49	C 23	ABUND	68.68	1.73	0.93	0.37	0.28
Q15.72	14.49	C 22	SPHERE	68.85	1.90	1.10	0.54	0.45
Q16.72	14.58	C 21	VOINIC	69.48	2.53	1.73	1.17	1.08
Q17.72	14.58	C 20	SOFRU	69.53	2.58	1.78	1.22	1.13
Q18.72	14.66	C 19	BASILIO	70.34	3.39	2.59	2.03	1.94
Q19.72	14.66	C 18	SOSTHENE	70.36	3.41	2.61	2.05	1.96
Q20.72	14.75	C 17	GABRIO	70.61	3.66	2.86	2.30	2.21
Q21.72	14.75	C 16	APEXUS	70.67	3.72	2.92	2.36	2.27
Q22.72	15.05	C 15	URSITA	71.26	4.31	3.51	2.95	2.86
Q23.72	15.05	C 14	GLOSA	72.19	5.24	4.44	3.88	3.79
Q24.72	15.05	C 13	SOTHYS	72.32	5.37	4.57	4.01	3.92
Q25.72	15.05	C 12	SOLINDO	72.39	5.44	4.64	4.08	3.99
Q26.72	15.05	C 11	CERTIVA	72.39	5.44	4.64	4.08	3.99
Q27.72	15.05	C 10	SOPHIE	72.45	5.50	4.70	4.14	4.05
Q28.72	15.05	C 9	CIPRIAN	72.71	5.76	4.96	4.40	4.31
Q29.72	15.05	C 8	CHEVIGNON	72.83	5.88	5.08	4.52	4.43
Q30.72	15.05	C 7	LITERA	75.07	8.12	7.32	6.76	6.67
Q31.72	15.05	C 6	CENTURION	76.66	9.71	8.91	8.35	8.26
Q32.72	15.05	C 5	VIVENDO	78.19	11.24	10.44	9.88	9.79
Q33.72	15.05	C 4	CRISANA	78.81	11.86	11.06	10.50	10.41
Q34.72	15.05	C 3	ACTIVUS	79.07	12.12	11.32	10.76	10.67
Q35.72	15.05	C 2	RUBISKO	80.26	13.31	12.51	11.95	11.86
Q36.72	15.05	C 1	SACRAMENTO	84.18	17.23	16.43	15.87	15.78
Sample	Sample value	Control	Wheat varieties	Q/ha	AURELIUS	ABUND	SPHERE	
		C 36	DACIC	55.08				
Q2.72	12.03	C 35	PITAR	58.04				
Q3.72	12.67	C 34	ASPEKT	59.48				
Q4.72	13.05	C 33	MIRANDA	61.10				
Q5.72	13.35	C 32	COMBIN	61.52				
Q6.72	13.60	C 31	ALEX	64.37				
Q7.72	13.77	C 30	BIHARIA	66.26				
Q8.72	13.94	C 29	BOEMA	66.44				
Q9.72	14.07	C 28	TIKATAKA	66.95				
Q10.72	14.15	C 27	PAPILLON	67.75				
Q11.72	14.15	C 26	AREZZO	68.31				
Q12.72	14.32	C 25	TIBERIUS	68.40				
Q13.72	14.32	C 24	AURELIUS	68.55				
Q14.72	14.49	C 23	ABUND	68.68	0.13			
Q15.72	14.49	C 22	SPHERE	68.85	0.30	0.17		
Q16.72	14.58	C 21	VOINIC	69.48	0.93	0.80	0.63	
Q17.72	14.58	C 20	SOFRU	69.53	0.98	0.85	0.68	
Q18.72	14.66	C 19	BASILIO	70.34	1.79	1.66	1.49	
Q19.72	14.66	C 18	SOSTHENE	70.36	1.81	1.68	1.51	
Q20.72	14.75	C 17	GABRIO	70.61	2.06	1.93	1.76	
Q21.72	14.75	C 16	APEXUS	70.67	2.12	1.99	1.82	
Q22.72	15.05	C 15	URSITA	71.26	2.71	2.58	2.41	
Q23.72	15.05	C 14	GLOSA	72.19	3.64	3.51	3.34	
Q24.72	15.05	C 13	SOTHYS	72.32	3.77	3.64	3.47	
Q25.72	15.05	C 12	SOLINDO	72.39	3.84	3.71	3.54	
Q26.72	15.05	C 11	CERTIVA	72.39	3.84	3.71	3.54	
Q27.72	15.05	C 10	SOPHIE	72.45	3.90	3.77	3.60	
Q28.72	15.05	C 9	CIPRIAN	72.71	4.16	4.03	3.86	
Q29.72	15.05	C 8	CHEVIGNON	72.83	4.28	4.15	3.98	
Q30.72	15.05	C 7	LITERA	75.07	6.52	6.39	6.22	
Q31.72	15.05	C 6	CENTURION	76.66	8.11	7.98	7.81	
Q32.72	15.05	C 5	VIVENDO	78.19	9.64	9.51	9.34	
Q33.72	15.05	C 4	CRISANA	78.81	10.26	10.13	9.96	
Q34.72	15.05	C 3	ACTIVUS	79.07	10.52	10.39	10.22	
Q35.72	15.05	C 2	RUBISKO	80.26	11.71	11.58	11.41	
Q36.72	15.05	C 1	SACRAMENTO	84.18	15.63	15.50	15.33	

**Table 3 plants-14-03580-t003:** Technology applied to tested wheat cultivars.

n	2021–2022	2022–2023	2023–2024
Preceding crop	Pea	Rapeseed	Pea
Summer plowing	23 July 2021		
Autumn plowing		6 September 2022	28 August 2023
Disc harrowing	23 July 2021	8 September 2022	29 August 2023/7 September 2023
Disc harrowing	26 October 2021	30 September 2022	10 November 2023
Date of harvest	27 October 2021	6 October 2022	13 November 2023
Seed treatment	Redigo Pro (prothioconazole 100 g/L + tebuconazole 60 g/L) − 0.50 l/t	Celest Extra (fludioxonil 25 g/L + difenoconazol 25 g/L) − 1 l/t	Celest Extra (fludioxonil 25 g/L + difenoconazol 25 g/L) − 1 l/t
Sowing	1 November 2021	21 October 2022	15 November 2023
Irrigation		9 November 2022 (30 mm)	
Sprouting	19 November 2021	20 November 2022	19 December 2023
Fertilization	15 March 2022	6 March 2023	20 March 2024
Herbicide treatment	16 April 2022 Axial One (pinoxaden 45 g/L + cloquintocet-mexyl 11.25 g/L + florasulam 7.50 g/L) − 1 l/ha	13 April 2023 Andor (Tritosulfuron 714 g/kg + Florasulam 54 g/kg) − 0.07 l/ha + Dash (methylated rapeseed oil − adjuvant, 100%) − 1 l/ha	30 March 2024 Omnera (fluroxypyr 135 g/L + thifensulfuron metil 30 g/L + metsulfuron metil 5 g/L) − 1 l/ha
Treatment 1	3 May 2022 Amistar (azoxystrobin 250 g/L) − 0.75 l/ha + Celsivo (fenpropidin 750 g/L) − 0.40 l/ha + Karate Zeon (lambda-cyhalothrin 50 g/L) − 0.15 l/ha	19 April 2023 Amistar (azoxystrobin 250 g/L) − 0.75 l/ha + Celsivo (fenpropidin 750 g/L) − 0.40 l/ha	1 April 2024 Amistar (azoxystrobin 250 g/L) − 0.75 l/ha + Celsivo (fenpropidin 750 g/L) − 0.40 l/ha + Karate Zeon (lambda-cyhalothrin 50 g/L) − 0.15 l/ha
Treatment 2	13 May 2022 Elatus Era (benzovindiflupyr 75 g/L + prothioconazole 150 g/L) − 1 l/ha + Decis Expert (deltamethrin 100 g/L) − 0.06 l/ha	13 May 2023 Elatus Plus (benzovindiflupyr 100 g/L) − 0.75 l/ha + Rivior (tetraconazol 125 g/L) − 0.90 l/ha + Nexide (gamma-cyhalothrin 60 g/L) − 0.08 l/ha	10 May 2024 Falcon (protioconazol 53 g/L + spiroxamină 224 g/L + tebuconazol 148 g/L) − 0.80 l/ha + Decis Expert (deltame-thrin 100 g/L) − 0.08 l/ha
Treatment 3	16 June 2022 Krima (acetamiprid 200 g/kg) − 0.10 kg/ha		
Harvested	15 July 2022	11 July 2023	24 June 2024

## Data Availability

All the relevant data are presented in the manuscript.

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
