# Peer review of "Nitrogen Management of Variety Screening of Winter Wheat Cultivated on Chernozem Soil for Yield Optimization"

_plants, 2025, doi:10.3390/plants14233580_

Round 1

Reviewer 1 Report

Comments and Suggestions for Authors
  1. The title could be more specific, for instance, by highlighting "variety screening" or the "interaction between nitrogen form and cultivar".

  2. Abstract: The core conclusions need to be stated more clearly with quantitative results. For example: "Sole application of ammonium nitrogen significantly reduced yield by XX% (from XX to XX), while the nitrate+ammonium combination showed a non-significant yield increase compared to nitrate nitrogen alone (+XX%, P > 0.05)". This enhances the immediacy of the conclusions.

  3. In the Introduction, the phrase "the intensification of agricultural activities over the past decade has also generated several negative environmental consequences" could be simplified to "agricultural intensification has led to environmental issues such as climate change and resource depletion", avoiding repetition (as "climate change, resource depletion" are mentioned previously).

  4. Introduction: The distinction and innovation of this study compared to existing research should be more explicitly stated. For example, systematic comparisons of multiple nitrogen forms and cultivar interactions on Chernozem soil are relatively scarce.

  5. The Materials and Methods section does not mention the number of field experiment replicates (e.g., three replicates) or the details of the randomized block design. These should be supplemented to ensure the reliability of the statistical analysis (e.g., the rationality of error calculation).

  6. The initial available nitrogen, phosphorus, and potassium content, as well as the pH of the Chernozem soil, are crucial for nitrogen availability. The physicochemical properties of the soil before the experiment (e.g., nitrate-N and ammonium-N content in the 0-20 cm layer) should be determined and reported. This helps explain why higher nitrogen application rates (150/170 kg/ha) led to yield reduction (e.g., whether it was due to luxury uptake or stress induced by high baseline nitrogen levels).

  7. Standardize the notation for statistical significance: In Table 1, the "Semnification" column uses symbols like "ooo" and "o". These should be unified to internationally recognized symbols: * (P < 0.05), ** (P < 0.01), *** (P < 0.001). A footnote should be added to the table explaining the significance levels corresponding to each symbol (e.g., "***: P < 0.001") to avoid ambiguity.

  8. The results show a significant yield decrease at the 150/170 kg/ha application rates. Potential reasons should be discussed with reference to the literature (e.g., increased nitrogen leaching, reduced root activity, higher pest/disease risk). If related indicators were not measured (e.g., plant nitrogen content, biomass allocation), this limitation can be acknowledged in the Discussion, suggesting future research include physiological indicators like nitrate reductase activity or chlorophyll content.

  9. In the text, "Q/Ha" should be changed to "q/ha" (hectare is lowercase). "a.s." (active substance) should be fully defined upon first use (e.g., "active substance"). "Semnification" should be corrected to "Significance".

  10. Units should preferably follow international conventions; for example, "kg/ha" is suggested to be changed to "kg·ha⁻¹".

  11. There are minor instances of less fluent language expression (e.g., "semnification" should be "significance"). A thorough language polish throughout the manuscript is recommended.

Author Response

Dear Reviewer,

First of all, we thank you for accepting the revision of our manuscript as well as for all the comments and suggestions provided. All changes in the manuscript are highlighted in yellow. The manuscript was also revised from the point of view of the English language.

Point 1. The title could be more specific, for instance, by highlighting "variety screening" or the "interaction between nitrogen form and cultivar".

Response 1: We appreciate your suggestion; we have changed the title.

Point 2. Abstract: The core conclusions need to be stated more clearly with quantitative results. For example: "Sole application of ammonium nitrogen significantly reduced yield by XX% (from XX to XX), while the nitrate+ammonium combination showed a non-significant yield increase compared to nitrate nitrogen alone (+XX%, P > 0.05)". This enhances the immediacy of the conclusions.

Response 2: Thank you for your suggestion, we have added this phrase to the text:,,Sole application of ammonium nitrogen significantly reduced yield by 3,34% (from 70,66 q/ha to 68,3 q/ha), while the nitrate+ammonium combination showed a non-significant yield increase compared to nitrate nitrogen alone (+0,65%, P > 0.05)".

Point 3. In the Introduction, the phrase "the intensification of agricultural activities over the past decade has also generated several negative environmental consequences" could be simplified to "agricultural intensification has led to environmental issues such as climate change and resource depletion", avoiding repetition (as "climate change, resource depletion" are mentioned previously).

Response 3: We have amended the text accordingly.

Point 4. Introduction: The distinction and innovation of this study compared to existing research should be more explicitly stated. For example, systematic comparisons of multiple nitrogen forms and cultivar interactions on Chernozem soil are relatively scarce.

Response 4: As your suggestion, we have added this phrase to the text: „The originality of the study lies in the comparisons made between several forms of nitrogen, as well as in the multiple comparisons between a large number of wheat cultivars of different origins. These are relatively rare”.

Point 5. The Materials and Methods section does not mention the number of field experiment replicates (e.g., three replicates) or the details of the randomized block design. These should be supplemented to ensure the reliability of the statistical analysis (e.g., the rationality of error calculation).

Response 5. We have amended the text accordingly. The experiment was conducted in subdivided plots with three replicates for each of the three-factor interactions: cultivar x nitrogen form x nitrogen dose. The cultivars represented the large plots, the nitrogen forms represented the medium plots, and the doses represented the small plots. First, the variability of the totals of the large plots occupied by factor A was analyzed, which was determined by the action of factor A, the differences between replicates, and the random fluctuation corresponding to the large plots. Then, the overall variability of the medium plots occupied by factor B was analyzed, which also includes the influence of factor B, the interaction A x B, and the random fluctuation corresponding to the medium plots. Lastly, the variability of the small plots was studied, which in fact represents the total variability of the experiment.

Point 6. The initial available nitrogen, phosphorus, and potassium content, as well as the pH of the Chernozem soil, are crucial for nitrogen availability. The physicochemical properties of the soil before the experiment (e.g., nitrate-N and ammonium-N content in the 0-20 cm layer) should be determined and reported. This helps explain why higher nitrogen application rates (150/170 kg·ha⁻¹) led to yield reduction (e.g., whether it was due to luxury uptake or stress induced by high baseline nitrogen levels).  

Response 6. We have amended the text accordingly. The soil shows a humus content of 3.04% in the Ap horizon and 1.78% in the Am horizon, decreasing by 1.24% in the B horizon. It has a medium total nitrogen supply, with 0.172% in Ap, a very good supply of mobile phosphorus (168 ppm) in Ap, and a good supply of mobile potassium (248 ppm). The soil reaction on the profile is slightly acidic to slightly alkaline. The sum of exchangeable bases is medium (23.2 in Ap, 24.8 in Am), and the degree of base saturation is 92.62%, gradually increasing in the Cca horizon. The microelement content is high for zinc (176 ppm) in Ap and extremely low in the B horizon (0,40 ppm).

Point 7. Standardize the notation for statistical significance: In Table 1, the "Semnification" column uses symbols like "ooo" and "o". These should be unified to internationally recognized symbols: * (P < 0.05), ** (P < 0.01), *** (P < 0.001). A footnote should be added to the table explaining the significance levels corresponding to each symbol (e.g., "***: P < 0.001") to avoid ambiguity.

Response 7. Thank you for your observation. We have modified it.

Point 8. The results show a significant yield decrease at the 150/170 kg·ha⁻¹ application rates. Potential reasons should be discussed with reference to the literature (e.g., increased nitrogen leaching, reduced root activity, higher pest/disease risk). If related indicators were not measured (e.g., plant nitrogen content, biomass allocation), this limitation can be acknowledged in the Discussion, suggesting future research include physiological indicators like nitrate reductase activity or chlorophyll content.

Response 8: We have amended the text accordingly. Future research will focus on the inclusion of physiological indicators such as chlorophyll content. The lack of these indicators, as well as the lack of determination of plant nitrogen content or biomass allocation, represents a certain limitation of this study.

Point 9. In the text, "Q/Ha" should be changed to "q/ha" (hectare is lowercase). "a.s." (active substance) should be fully defined upon first use (e.g., "active substance"). "Semnification" should be corrected to "Significance".

Response 9: Thank you for your observation. However, we will use Q/ha and Q/HA in the figures because changing them requires more time, which is unfortunately limited in order to respond to the three reviewers within the set time frame.

Point 10. Units should preferably follow international conventions; for example, "kg·ha⁻¹" is suggested to be changed to "kg·ha⁻¹.

Response 10: Thank you for your observation. We have modified it.

Point 11. There are minor instances of less fluent language expression (e.g., "semnification" should be "significance"). A thorough language polish throughout the manuscript is recommended.

Response 11: Thank you for your observation. We have modified it.

Thank you for your time and valuable suggestions.

Yours faithfully,

The Authors

Reviewer 2 Report

Comments and Suggestions for Authors

General Comments:

The manuscript requires major revision before it can be considered for publication. The Introduction contains repetitive sentences and short paragraphs that should be merged for better flow. The methods section is unclear please specify the number of field replications and describe all management practices other than nitrogen fertilizer rates. All other sections needs to be revised and rewritten and avoid short paragraphs.

Introduction:
The manuscript is not well written and requires substantial revision before it can be considered for publication. There is considerable repetition throughout the text, especially in the Introduction. Many sentences convey the same meaning, and several short paragraphs could be merged to improve flow and readability.

Results and Discussion:
All figure and table captions should be rewritten clearly to accurately reflect the analyses performed. The captions should include information such as the number of replicates and any relevant experimental details. Moreover, all figures currently lack standard deviation or standard error bars, making it difficult for readers to assess the variability and reliability of the results. For example, it is not clear how many replicates were included for each wheat variety.
Lines 131 - 148 should be rewritten in paragraph form rather than as bullet points, as this will improve coherence and readability.
In Figure 1, please clarify the terminology - nitric N and nitrate N are not the same. Use the correct term consistently throughout the manuscript.

Methods:
The Methods section lacks essential details. It is unclear how many field replicates were used and what management practices, aside from nitrogen fertilizer rates, were applied. These details are critical for reproducibility and interpretation of the results, and they must be clearly stated.

Comments on the Quality of English Language

Need minor correction

Author Response

Dear Reviewer,

First of all, we thank you for accepting the revision of our manuscript as well as for all the comments and suggestions provided. All changes in the manuscript are highlighted in yellow. The manuscript was also revised from the point of view of the English language.

Point 1. General Comments:

The manuscript requires major revision before it can be considered for publication. The Introduction contains repetitive sentences and short paragraphs that should be merged for better flow. The methods section is unclear please specify the number of field replications and describe all management practices other than nitrogen fertilizer rates. All other sections needs to be revised and rewritten and avoid short paragraphs.

Response 1: We inserted the number of repetitions and the agricultural technologies applied.

Point 2. Introduction:

The manuscript is not well written and requires substantial revision before it can be considered for publication. There is considerable repetition throughout the text, especially in the Introduction. Many sentences convey the same meaning, and several short paragraphs could be merged to improve flow and readability.

Response 2: We have followed your suggestions, and the introduction has been rearranged.

Point. 3. Results and Discussion:

All figure and table captions should be rewritten clearly to accurately reflect the analyses performed. The captions should include information such as the number of replicates and any relevant experimental details

Response 3. We have filled in the number of repetitions and P values in the Results and Discussion section.

Point 4. Moreover, all figures currently lack standard deviation or standard error bars, making it difficult for readers to assess the variability and reliability of the results. For example, it is not clear how many replicates were included for each wheat variety.

Response 4. We chose to present the deviations in a much more visible way than standard deviation bars. The meanings represented by circles and stars based on the values: * (P < 0.05), ** (P < 0.01), *** (P < 0.001) are frequently used in the works of Romanian researchers.

Point 5. Lines 131 - 148 should be rewritten in paragraph form rather than as bullet points, as this will improve coherence and readability.

Response 5. Thank you for your observation. We have modified it.

Point 6. In Figure 1, please clarify the terminology - nitric N and nitrate N are not the same. Use the correct term consistently throughout the manuscript.

Response 6. Good observation, thank you. We have amended the entire manuscript.

Point 7. Methods:

The Methods section lacks essential details. It is unclear how many field replicates were used and what management practices, aside from nitrogen fertilizer rates, were applied. These details are critical for reproducibility and interpretation of the results, and they must be clearly stated.

Response 7. We have introduced the number of repetitions and the agricultural technologies applied.

Thank you for your time and valuable suggestions.

Yours faithfully,

The Authors

Reviewer 3 Report

Comments and Suggestions for Authors

This is an important and valuable study because it was to identify the nitrogen form that significantly influences wheat yield, as well as the cultivars that respond positively to specific form of nitrogen fertilization.

However, I think significant modifications are needed. 

Abstract

L25 Is “three years (2021–2024)” meant to be “three seasons (2021–2024)”?

L35 “both controls” — It’s unclear what “controls” refers to. I think an explanation is needed.

  1. Introduction

There are many instances where a single paragraph consists of just one sentence, such as in L105-107, L119-122, and L123-124; typically, a paragraph should consist of multiple sentences.

L105-107 This paragraph doesn't seem to have much connection with the preceding and following paragraphs. I think it would be better to place it in the first half of the Introduction.

Since the purpose of this study is not described, please include it at the end of the Introduction.

  1. Results and Discussion

L141 Isn’t “Aspeckt” supposed to be “Aspekt”?

L143 From the results in Table 1, shouldn't "Tiberius" be included with "Papillon, Boema, Arezzo"?   “Papillon, Boema, Tiberius, Arezzo”

Table 1

It is necessary to explain what the yellow and pink shading indicate.

Is “Semnification” supposed to be “Signification”?

Explanations for the symbols such as asterisks (*, **, ***, º, ºº, ººº) are needed.

L161-166 “Since weather conditions vary from year to year, I think it would be easier to understand if you also showed yield stability across years (such as standard error or coefficient of variation).

Figure 1

An explanation is needed for the symbol “º” attached to the value “68.3” for “ammonium N.”

In the figure, shouldn’t “amonium N” be “ammonium N”?

Shouldn’t the unit for yield shown as “Q/HA” in the figure be “Q/ha”? I think it’s better to unify it with the unit used in the main text.

Figure 2

An explanation of “first dose,” “second dose,” and “third dose” in the figure is needed in a footnote.

Shouldn’t the unit for yield “q/ha” in the figure be “Q/ha”? I think it’s better to unify it with the unit used in the main text.

An explanation in a footnote is needed for the “º” symbol attached to the values “69.6” and “69.42” in the figure.

Figures 3-10 

Because the numbers and letters on the vertical and horizontal axes of the figure are small, they are hard to see. Please use a slightly larger size.  The text in the legend is also quite small, so please increase the size a bit.

Regarding the variety names, in the main text and figure titles, only the first letter is capitalized (for example, "Glosa"), but in the figures, all letters are capitalized (for example, "GLOSA"). It is better to use the official cultivar name notation. This also applies to other figures and tables. 

Should the unit for yield in the figure, "Q/HA," be "Q/ha"? I think it would be better to unify it with the units used in the main text. 

An explanation is needed in each figure's footnote for the symbols "*" and "º" attached to the numbers in the figures, as well as for the numbers shown in bold. 

In some figures, under the row "difC-Ct2," there is a notation such as "P>5%  5.64" (only in Figure 6(b), Figure 10(a), and (b)), but adding an explanation in the footnote of each figure should suffice.

L313 "Tika Taka" should be "Tika-Taka." 

L314 Regarding "Abund," looking at the results in Figure 9(c), the values for ammonium nitrogen application tend to be higher than those for Ct1 and Ct2, so I think you need to remove it from the statement in L314. 

Figure 11

The text on the vertical and horizontal axes is easier to see in "black" rather than "gray."

L362 "Tika Taka" should be "Tika-Taka."

As for unit notation, should it be "kg/ha" or "kg ha^-1"? Since both are mixed in the main text, use the correct notation specified by the journal.

Table 2 

An explanation is needed for what the green shading in the table indicates.

  1. Materials and Methods

L388 Is “three years (2022–2024)” meant to be “three seasons (2021–2024)”? Expression consistency with “three years (2021–2024)” in L25 is necessary.

L396-400 The objective section should be included in the Introduction.

L403 For the “Chernozem soil”, more detailed data or explanations about its chemical properties (such as pH, soil nutrients, especially nitrogen content, etc.) are required.

There is no detailed description of the cultivation methods. Please specify when and how sowing was done each year (seedbed preparation, sowing, etc.), how fertilization was carried out (basal dressing or top dressing, whether it was applied separately, etc.), management during the cultivation period (irrigation, weed control, etc.), as well as details on sampling and harvesting methods.

L409 Please specify the source of the “Meteorological data”.

L428 Should “P > 5%” be “P < 0.05”?

L430-431 There are many cases where only one sentence constitutes a paragraph; usually, a paragraph consists of multiple sentences.

L430-435 Information regarding the software used for statistical analysis needs to be included.

Author Response

Dear Reviewer,

First of all, we thank you for accepting the revision of our manuscript as well as for all the comments and suggestions provided. All changes in the manuscript are highlighted in yellow. The manuscript was also revised from the point of view of the English language.

Point 1. This is an important and valuable study because it was to identify the nitrogen form that significantly influences wheat yield, as well as the cultivars that respond positively to specific form of nitrogen fertilization. However, I think significant modifications are needed.

Abstract

L25 Is “three years (2021–2024)” meant to be “three seasons (2021–2024)”?

Response 1: Three agricultural years (2021-2022, 2022-2023, 2023-2024). We have amended the text accordingly.

Point 2. L35 “both controls” — It’s unclear what “controls” refers to. I think an explanation is needed.

Response 2: This refers to the two witnesses (control or check in English).  We kept control, abbreviated Ct. We can modify it as follows: both in relation to Ct1– the most commonly grown wheat cultivar Glosa and in relation to control 2 (Ct2) – the average yield of all tested cultivars.  This was also explained in the manuscript.

Point 3. Introduction

There are many instances where a single paragraph consists of just one sentence, such as in L105-107, L119-122, and L123-124; typically, a paragraph should consist of multiple sentences.

Response 3: We have rearranged the text according to your suggestion.

Point 4. L105-107 This paragraph doesn't seem to have much connection with the preceding and following paragraphs. I think it would be better to place it in the first half of the Introduction.

Response 4: We have taken the recommendation into account.

Point 5. Since the purpose of this study is not described, please include it at the end of the Introduction.

Response 5. We introduce the purpose at the end of the introduction: The aim of the study was to identify the nitrogen form that significantly influences wheat yield, as well as the cultivars that respond positively to specific form of nitrogen fertilization, in order to provide recommendations regarding cultivar selection and the appropriate technological approach for chernozem soils in southern Romania.

Point 6. Results and Discussion

L141 Isn’t “Aspeckt” supposed to be “Aspekt”?

Response 6. The correct form is indeed Aspekt and we have modified that accordingly now.

Point 7. L143 From the results in Table 1, shouldn't "Tiberius" be included with "Papillon, Boema, Arezzo"?   “Papillon, Boema, Tiberius, Arezzo”.

Response 7. Yes, thank you for the good observation. We have also introduced Tiberius.

Point 8. Table 1

It is necessary to explain what the yellow and pink shading indicate.

Response 8: Values marked in yellow represent statistically significant increases. Values marked in pink represent statistically significant decreases. This was also mentioned in the manuscript.

Point 9. Is “Semnification” supposed to be “Signification”?

Response 9: Good observation, we have modified it accordignly.

Point 10. Explanations for the symbols such as asterisks (*, **, ***, º, ºº, ººº) are needed.

Response 10: We have explained the symbols based on your request:

  *, o (P < 0.05) = 3.74 Q/ha, **, oo (P < 0.01) = 5.31 Q/ha, ***, ooo (P < 0.001) = 7.69 Q/ha.

Point 11. L161-166 “Since weather conditions vary from year to year, I think it would be easier to understand if you also showed yield stability across years (such as standard error or coefficient of variation).

Response 11: The climatic conditions are presented in detail in L409-420. We consider that the tested varieties are highly differentiated in terms of vegetation period and the coefficient of variability calculated for each year is not conclusive. However, the coefficient of variability is calculated for the average values of the yields obtained during the three years.

Point 12. Figure 1

An explanation is needed for the symbol “º” attached to the value “68.3” for “ammonium N.

In the figure, shouldn’t “amonium N” be “ammonium N”?

Response 12. We have modified the figure as per your suggestion.

Point 13. Shouldn’t the unit for yield shown as “Q/HA” in the figure be “Q/ha”? I think it’s better to unify it with the unit used in the main text.

Response 13. We have amended it in the manuscript.

Point 14. Figure 2

An explanation of “first dose,” “second dose,” and “third dose” in the figure is needed in a footnote.

Response 14. We have mentioned the explanation below the figure: first dose: 120 kg/ha a.s., second dose: 150 kg/ha a.s., third dose: 170 kg/ha a.s.

Point 15. Shouldn’t the unit for yield “q/ha” in the figure be “Q/ha”? I think it’s better to unify it with the unit used in the main text.

Response 15. We have modified that in the figure accordingly.

Point 16. An explanation in a footnote is needed for the “º” symbol attached to the values “69.6” and “69.42” in the figure.

Response 16. We have explained it underneath the figure:  o (P < 0.05) = 1.41 Q/ha.

Point 17. Figures 3-10

Because the numbers and letters on the vertical and horizontal axes of the figure are small, they are hard to see. Please use a slightly larger size.  The text in the legend is also quite small, so please increase the size a bit.

Response 17. The figures have been enlarged.

Point 18. Regarding the variety names, in the main text and figure titles, only the first letter is capitalized (for example, "Glosa"), but in the figures, all letters are capitalized (for example, "GLOSA"). It is better to use the official cultivar name notation. This also applies to other figures and tables.

Response 18. Only the official names were used for all varieties. We do not consider it incorrect that they are written as Glosa or GLOSA.

Point 19. Should the unit for yield in the figure, "Q/HA," be "Q/ha"? I think it would be better to unify it with the units used in the main text.

Response 19. We have modified it in the text.

Point 20. An explanation is needed in each figure's footnote for the symbols "*" and "º" attached to the numbers in the figures, as well as for the numbers shown in bold.

Response 20. We have listed the explanation under each figure:   *, o (P < 0.05) = 5.64 Q/ha, **, oo (P < 0.01) = 8.35 Q/ha, ***, ooo (P < 0.001) = 11.52 Q/ha.

Point 21. In some figures, under the row "difC-Ct2," there is a notation such as "P>5% 5.64" (only in Figure 6(b), Figure 10(a), and (b)), but adding an explanation in the footnote of each figure should suffice.

Response 21. The respective notes have been removed.

Point 22. L313 "Tika Taka" should be "Tika-Taka."

Response 22. On the official website of Saaten Union, www.saaten-union.ro, the variety is listed under the name Tika Taka. We checked the entire manuscript.

Point 23. L314 Regarding "Abund," looking at the results in Figure 9(c), the values for ammonium nitrogen application tend to be higher than those for Ct1 and Ct2, so I think you need to remove it from the statement in L314.

Response 23. Good observation. We removed it.

Point 24. Figure 11

The text on the vertical and horizontal axes is easier to see in "black" rather than "gray."

Response 24. We have modified the colours, as per your suggestion.

Point 25. L362 "Tika Taka" should be "Tika-Taka."

Response 25. On the official website of Saaten Union, www.saaten-union.ro, the variety is listed under the name Tika Taka. We checked the entire manuscript.

Point 26. As for unit notation, should it be "kg/ha" or "kg ha^-1"? Since both are mixed in the main text, use the correct notation specified by the journal.

Response 26. We have modified kg/ha with kg·ha-1

Point 27. Table 2

An explanation is needed for what the green shading in the table indicates.

Response 27. The manuscript contained an explanation of the green highlighting.

Point 28. Materials and Methods

L388 Is “three years (2022–2024)” meant to be “three seasons (2021–2024)”? Expression consistency with “three years (2021–2024)” in L25 is necessary.

Response 28. Three agricultural years (2021-2022, 2022-2023, 2023-2024). We have amended it in the manuscript.

Point 29. L396-400 The objective section should be included in the Introduction.

Response 29. The objective of the paper was included in the introduction.

Point 30. L403 For the “Chernozem soil”, more detailed data or explanations about its chemical properties (such as pH, soil nutrients, especially nitrogen content, etc.) are required.

Response 30. Data on the physical and chemical properties of the soil were included in the chapter Material and Method.

Point 31. There is no detailed description of the cultivation methods. Please specify when and how sowing was done each year (seedbed preparation, sowing, etc.), how fertilization was carried out (basal dressing or top dressing, whether it was applied separately, etc.), management during the cultivation period (irrigation, weed control, etc.), as well as details on sampling and harvesting methods.

Response 31. The technological data were now presented in the chapter Material and Method.

Point 32. L409 Please specify the source of the “Meteorological data”.

Response 32. We have specified the source for the meteorological data.

Point 33. L428 Should “P > 5%” be “P < 0.05”?

Response 33. Good observation. It has been replaced. 

Point 34. L430-431 There are many cases where only one sentence constitutes a paragraph; usually, a paragraph consists of multiple sentences.

Response 34. Your request has been complied with.

Point 35. L430-435 Information regarding the software used for statistical analysis needs to be included.

Response 35. Information regarding the statistical analysis used has been inserted into the manuscript.

Thank you for your time and valuable suggestions.

Yours faithfully,

The Authors

Round 2

Reviewer 1 Report

Comments and Suggestions for Authors

The author revised the manuscript well and answered the reviewers' comments carefully. However, the title of the manuscript only needs to be capitalized.

Author Response

Response to Reviewer 1 Comments

Dear Reviewer,

Please allow us to thank you for accepting the revision of our manuscript, as well as for all the valuable comments and suggestions provided. In the latest revised version of the manuscript, all changes are highlighted in blue. The manuscript was also thoroughly proofread and checked by an authorized translator.

Point 1. The author revised the manuscript well and answered the reviewers' comments carefully. However, the title of the manuscript only needs to be capitalized.

Response 1: Thank you for your observation. The title of the manuscript is written in capital letters.

Thank you for your time and valuable suggestions.

Yours faithfully,

The Authors

Reviewer 2 Report

Comments and Suggestions for Authors

 In Figure 1, I previously suggested changing ‘nitric N’ to ‘nitrate N,’ but this correction has not been made.

Moreover, although  information about three replicates have been added in the methodology section, the standard errors (SE) of the means are not provided in the results, making it difficult to interpret the statistical validity of the findings. 

Author Response

Response to Reviewer 2 Comments

Dear Reviewer,

Please allow us to thank you for accepting the revision of our manuscript, as well as for all the valuable comments and suggestions provided. In the latest revised version of the manuscript, all changes are highlighted in blue. The manuscript was also thoroughly proofread and checked by an authorized translator.

Point 1. In Figure 1, I previously suggested changing ‘nitric N’ to ‘nitrate N,’ but this correction has not been made.

Response 1: Thank you very much for your observation. However, following your previously suggestion, we chose to replace the term “nitrate” with “nitric” throughout the entire manuscript. Therefore, we believe that the same terminology should be maintained in Figure 1 as well. Nitric nitrogen is the more precise and commonly used term in scientific and technical contexts, as it refers to the nitrate ion as a chemical form.

Point 2. Moreover, although information about three replicates have been added in the methodology section, the standard errors (SE) of the means are not provided in the results, making it difficult to interpret the statistical validity of the findings.

Response 2: In the revised version of the manuscript, we have completed as follows: Lines 148-150; Lines 178-184.

Thank you for your time and valuable suggestions.

Yours faithfully,

The Authors

Reviewer 3 Report

Comments and Suggestions for Authors

Thank you for responding to my comments. I have reviewed the revised sections. I feel that the manuscript has been improved and is now easier to understand. However, I believe there are still a few areas that require further revision. Please consider the following points.

I think it's better to standardize the number of decimal places for the numerical values in the Figures and Tables.

For example: "68.3" → "68.30", "59" → "59.00", etc.

There is a mixture of periods (".") and commas (",") used as decimal points in the text and Tables.

For example: "70,66" → "70.66", "0,65" → "0.65", etc.

Since this journal uses periods (".") as decimal points, please revise them accordingly.

Regarding Figure 3-10

Explanations for the symbols "*" and "º" attached to the numbers in the Figure have been added, but, for example, in Figure 3, neither "*" nor "º" are attached to any numbers in the figure, so isn't this unnecessary?

In Figures 4-6, only the symbol "*" is attached to numbers, so wouldn't "* (P < 0.05) = 5.64 Q/ha" alone be sufficient?

For Figures 7 and 8, only "o (P < 0.05) = 5.64 Q/ha" would be sufficient; for Figures 9 and 10, wouldn't "* , o (P < 0.05) = 5.64 Q/ha" alone suffice?

Table 3

Does "Preparatory" in the table mean "Preceding crop"?

Does "Combine harvester" in the table refer to "date of harvest"? Or "land preparation"?

L 433-434

I think additional explanation is necessary for soil terms such as "Ap," "Am," "B," and "Cca."

Author Response

Response to Reviewer 3 Comments

Dear Reviewer,

Please allow us to thank you for accepting the revision of our manuscript, as well as for all the valuable comments and suggestions provided. In the latest revised version of the manuscript, all changes are highlighted in blue, except only for the numbers from figures and tables. The manuscript was also thoroughly proofread and checked by an authorized translator.

Point 1. Thank you for responding to my comments. I have reviewed the revised sections. I feel that the manuscript has been improved and is now easier to understand. However, I believe there are still a few areas that require further revision. Please consider the following points.

Response 1: Thank you, we have taken the recommendation into account.

Point 2. I think it's better to standardize the number of decimal places for the numerical values in the Figures and Tables. For example: "68.3" → "68.30", "59" → "59.00", etc.

Response 2: We have made all corrections.

Point 3. There is a mixture of periods (".") and commas (",") used as decimal points in the text and Tables. For example: "70,66" → "70.66", "0,65" → "0.65", etc. Since this journal uses periods (".") as decimal points, please revise them accordingly.

Response 3: We have made all corrections.

Point 4. Regarding Figure 3-10. Explanations for the symbols "*" and "º" attached to the numbers in the Figure have been added, but, for example, in Figure 3, neither "*" nor "º" are attached to any numbers in the figure, so isn't this unnecessary?

Response 4: Your observations are correct. We have made the corrections.

Point 5. In Figures 4-6, only the symbol "*" is attached to numbers, so wouldn't "* (P < 0.05) = 5.64 Q/ha" alone be sufficient?

Response 5. Your observations are correct. We have made the corrections.

Point 6. For Figures 7 and 8, only "o (P < 0.05) = 5.64 Q/ha" would be sufficient; for Figures 9 and 10, wouldn't "* , o (P < 0.05) = 5.64 Q/ha" alone suffice?

Response 6. Your observations are correct. We have made the corrections.

Point 7. Table 3. Does "Preparatory" in the table mean "Preceding crop"? Does "Combine harvester" in the table refer to "date of harvest"? Or "land preparation"?

Response 7. Preceding crop is correct. Also, date of harvest is correct. We have made the corrections.

Point 8. L 433-434. I think additional explanation is necessary for soil terms such as "Ap," "Am," "B," and "Cca."

Response 8: Your observations are correct. We have added the necessary explanations.

Thank you for your time and valuable suggestions.

Yours faithfully,

The Authors
